# Profile of Bioactive Components of Cocoa (*Theobroma cacao* L.) By-Products from Ecuador and Evaluation of Their Antioxidant Activity

**DOI:** 10.3390/foods12132583

**Published:** 2023-07-03

**Authors:** Wilma Llerena, Iván Samaniego, Christian Vallejo, Adner Arreaga, Billy Zhunio, Zomayra Coronel, James Quiroz, Ignacio Angós, Wilman Carrillo

**Affiliations:** 1Facultad de Ciencia de la Industria y la Producción, Universidad Técnica Estatal de Quevedo (UTEQ), km 7 1/2 vía Quevedo-El Empalme, Quevedo 120301, Ecuador; wllerenas@uteq.edu.ec (W.L.); cvallejo@uteq.edu.ec (C.V.); adner.arreaga@uteq.edu.ec (A.A.); billy.zhunio2015@uteq.edu.ec (B.Z.); zomayra.coronel2015@uteq.edu.ec (Z.C.); 2Departamento de Nutrición y Calidad, Instituto Nacional de Investigaciones Agropecuarias (INIAP), Panamericana Sur km 1, Cutuglahua 171107, Ecuador; ivan.samaniego@iniap.gob.ec; 3Programa de Cacao, Instituto Nacional de Investigaciones Agropecuarias (INIAP), Litoral Sur Experimental Station, km 26 via Duran-El Tambo, Yaguachi 092406, Ecuador; james.quiroz@iniap.gob.ec; 4Departamento de Agronomía, Biotecnología y Alimentación, Universidad Pública de Navarra (UPNA), 31006 Pamplona, Spain; ignacio.angos@unavarra.es; 5Facultad de Ciencia e Ingeniería en Alimentos y Biotecnología, Universidad Técnica de Ambato (UTA), Av. Los Chasquis y Río Payamino, Ambato 180103, Ecuador

**Keywords:** cacao, Nacional X Trinitario type, CCN-51, mucilage, bean shell, polyphenols, flavonoids, antioxidant activity

## Abstract

The aim of the study was to determine the profile of bioactive compounds in cocoa residues (mucilage and bean shells), and to evaluate their antioxidant activity in two cocoa varieties, Nacional X Trinitario type (Fine Aroma) and the variety CCN-51. The extraction of phytonutrients from the residues was carried out selectively. The characterization and quantification of the total polyphenol content (TPC), and the total flavonoid content (TFC) were determined by UV–VIS spectrophotometry. High-performance liquid chromatography (HPLC) was used to determine the phenolic profile and methylxanthines. The antioxidant activity was evaluated by the methods of 2-azinobis (3-ethyl-benzothiazoline-6-sulfonic acid) cation bleaching (ABTS), ferric-reducing antioxidant power (FRAP) and oxygen radical absorbance capacity (ORAC). The exudate mucilage samples from Nacional X Trinitario-type cocoa presented the highest content of TPC 105.08 mg gallic acid equivalents (GAE)/100 mL, TFC 36.80 mg catechin equivalents (CE)/100 mL, catechin (CAT) 35.44 mg/g, procyanidins (PCB2: 35.10; PCB1: 25.68; PCC1: 16.83 mg/L), epicatechin (EPI) 13.71 mg/L, caffeine (CAF) 0.90% and theobromine (TBR) 2.65%. In the cocoa bean shell, the variety CCN-51 presented a higher content of TPC (42.17 mg GAE/100 g) and TFC (20.57 mg CE/100 g). However, CAT (16.16 mg/g), CAF (0.35%) and TBR (1.28%) were higher in the Nacional X Trinitario cocoa type. The EPI presented no significant differences between the two samples studied (0.83 and 0.84 mg/g). The antioxidant activity values (ABTS, FRAP and ORAC methods) were higher in the samples of CCN-51 than in the Nacional X Trinitario type. The bean shell samples presented antioxidant values of 171.32, 192.22 and 56.87 mg Trolox equivalents (TE)/g, respectively, and the bean shell samples presented antioxidant values of 167.06, 160.06 and 52.53 mg TE/g, respectively. The antioxidant activity (ABTS, FRAP and ORAC) of the residues was correlated with the bioactive compounds of the mucilage and bean shells, showing a strong positive correlation (<0.99) with the procyanidins (B1, B2 and C1), EPI and CAT and a positive/moderate correlation (0.94) with methylxanthines.

## 1. Introduction

Ecuador is an agricultural country where the agricultural sector contributes 8.25% of real GDP (gross domestic product). In 2020, the surface area of permanent crops reached 1,504,694 hectares, distributed between different crops: cocoa (39.25%), African palm (17.07%), banana (10.97%), plantain (9.67%), sugar cane (9.44%), and other permanent crops (13.60%) [1,2].

According to the Food and Agriculture Organization of the United Nations (FAO), ref. [3] cocoa production is considered one of the most important items in the country, ranking first in planted area (590,579 ha), harvested area (527,347 ha), production (327,903 t) and yield (0.62 ton/ha). It is followed by other crops such as hard corn, rice, palm oil, banana, plantain, sugar cane, soft corn, coffee, potato, among others. The province with the highest production volume is Los Ríos (26.68%), followed by Guayas (19.57%), Manabí (19.10%), Esmeraldas (13.96%) and Santo Domingo de Los Tsáchilas (4.79%) on the Pacific coast. The rest of the provinces of the coast and Amazon reach percentages around 20% of the total [1].

By 2022, in Ecuador, cocoa reached an agricultural participation of 11.10% with 28,094 t and an income of more than USD 71 million from exports. The highest economic income corresponds to the sale of cocoa beans (USD 6,358,242.7 million) followed by cocoa liquor, chocolate, cocoa butter, cocoa powder, cocoa paste, cocoa shell, fat and cocoa oil [3].

Currently, there are between 101 and 155 types and 4 genotypes of *Theobroma cacao* L.; its cultivars are differentiated by the characteristics and morphological traits of the cob and grains. There are four commercial varieties called Criollo, Forastero, Trinitario (a hybrid of Criollo and Forastero) and the well-known Nacional de Ecuador [4,5].

In Ecuador, different types, cultivars and varieties of cocoa are grown, with the predominant clone of cocoa CCN-51 (Collection of Castro Naranjal) and cocoa of the Nacional X Trinitario type, also known as “Fino de Aroma” or “Arriba” [4]. The CCN-51 cocoa variety has high productivity, quality and tolerance to diseases; however, it is a type of bulk almond used to produce lower quality chocolates, mainly for confectionery uses, and products such as cocoa powder. The Nacional X Trinitario-type variety is highly demanded by chocolatiers, as it is used to produce the finest chocolates, thanks to its aroma and fruity and floral flavors [5,6]. The postharvest process and the industrialization of food generates large volumes of waste, with roots, tubers and oilseed crops (25%) producing the largest amount of biomass during their processing; these are followed by other production sectors such as fruit and vegetables (22%), meat and animal products (12%), cereals and legumes (9%) [7].

The European Commission defines food waste as residue, in a liquid or solid state with a high organic content, derived from food production and food processing. This residue is removed from production processing. Food by-products are defined as functional compounds, obtained from food waste, for the development of new products with a market value. The food by-products obtained from plant processing represent a low-cost alternative for the production of functional compounds. Cocoa by-products have been described as having a high content of polyphenol and flavonoid compounds with antioxidant activity. It is important to identify and characterize these components in the different varieties of cocoa. For this reason, this research proposes the characterization and identification of the phenolic compounds present in the cocoa by-products of two varieties of Ecuadorian cocoa and the antioxidant evaluation of the extracts [8,9,10].

The demand for cocoa increases every year, which is why for decades the cocoa industry has been affected by its sustainability. About 10 tons of wet waste and cocoa by-products are disposed of for every ton of dry beans, making waste management extremely complex and expensive [11]; in the cocoa benefit chain, the kernel constitutes less than one third (16–21%) of the weight of the fruit, leaving behind approximately two thirds of residual biomass between shell, placenta and mucilage exudate as waste by-products. These components of the fruit are generally discarded on the farm because they are considered garbage. They are used as organic fertilizers on farms, which allows the nutrients to be returned to the soil and recycled, after decomposition, in forms available to plants. However, the indiscriminate accumulation of waste, deposited in the soil, causes pollution to the environment and the proliferation of pests and disease [12,13,14].

Consumers are more concerned about the possible health benefits of their diet. This fact has increased the demand for foods that contain biologically active compounds. The new challenge for the food industry is to enrich its products for human consumption with compounds that have functional properties such as antioxidant compounds. Specifically, polyphenol compounds are of great interest due to their possible positive effects on human health, such as anticancer, antioxidant, and anti-inflammatory activity. Consequently, food science research is focused on finding new and sustainable sources of antioxidants, optimizing extraction and purification methods as well as developing innovative health-promoting functional foods [15,16,17]. 

It has been projected that, by 2050, the population of the planet will be around 9.8 billion people; it is necessary to use resources properly, recycle and reuse by-products. This population increase necessitates the search for new food sources [18]; the recovery of food waste and its by-products is an alternative for improving the economic and environmental sustainability of the food production chain, accelerating the transition of the food industry towards a circular bioeconomy, which would allow the goal of zero waste to be achieved [19,20,21,22]. In addition, it is one of the pillars of the circular economy to preserve the value of products, materials and resources for a long period, through the use and production of renewable energy from biological sources and obtaining products of added value (food and phytonutrients) from the resources [19].

The recovery of these high value-added compounds from agro-industrial waste is a topic of scientific interest, because the non-edible portions of agri-food products can contain high amounts of phytonutrients and valuable bioactive compounds [23,24].

At a scientific and industrial level, different waste management strategies are being evaluated to convert waste into derivatives of high biological and technological value such as food additives, nutraceuticals, phototherapeutics, and cosmetics, among others [23]. Sugars, pectin, proteins, lipids, polysaccharides, fibers, polyphenols, vitamins (A and E), essential minerals, fatty acids, volatiles, anthocyanins and pigments are generally recovered from fruits and vegetables [23].

Residues from the cocoa benefit chain are rich in bioactive molecules such as methylxanthines (TBR and CAF) and phenolic compounds (phenolic acids and flavonoids) that may provide benefits to human health [24]. The consumption of bioactive compounds in small concentrations has been shown to have antioxidant and anti-mutagenic properties. Due to the ability of these compounds to delay or prevent the oxidative degradation of other molecules (DNA, lipids, carbohydrates and proteins), they reduce the oxidative effect of free radicals; therefore, they reduce the risk of chronic diseases such as cancer, diabetes and cardiovascular pathologies [25].

The waste, to be used for the production of by-products, must meet a number of legal requirements in terms of the environment, safety and quality. Legislation and directives are critical factors that hinder the recovery of food by-products for use as food ingredients and natural additives [26]. Likewise, they must comply with the regulations for novel foods of each country [27]. 

The objective of this work was to determine the bioactive compound content in cocoa residues (mucilage and husk) of the varieties CCN-51 and Nacional X Trinitario type and the make an evaluation of the antioxidant activity of the by-products using the ABTS, FRAP and ORAC methods.

## 2. Materials and Methods

### 2.1. Plant Material

The sampling and collection of cocoa was carried out by randomly choosing cocoa trees of the National X Trinitario-type variety (Fino de Aroma) and the CCN-51 variety from the farms of producers of the “La Cruz” Association of the Mocache canton, province of Los Ríos in Ecuador. The cocoa production process was carried out using 100 kg of each variety, from which the residues used as study material (cocoa mucilage and bean shells) were obtained.

### 2.2. Cocoa Fermentation Process

The postharvest process of the collected cocoa samples was carried out in the fermentation and drying center of the “La Cruz” Association, located in the Mocache canton, province of Los Ríos, following the method proposed by Samaniego et al. [5]. The grains were extracted from each fruit and fermented in bay wood boxes (100 × 100 × 95 cm) (height × width × depth), with a capacity for 150 kg of dough, located in a staggered manner on three floors.

In Nacional X Trinitario-type cacao, fermentation was carried out for 4 days. Cocoa beans were placed for 2 days in the upper drawer, 1 day in the middle drawer and 1 day in the lower drawer. For CCN-51 cocoa, fermentation was carried out for 6 days with 2 days in each drawer. To ensure an adequate fermentation process, the grain was removed every 24 h and the dough was turned every 48 h.

During fermentation, the mucilage samples exuded or leached from the almonds were collected at the base of each box, in hermetic bags (Ziploc) isolated from light and oxygen. Subsequently, each sample was stored frozen at a temperature of −20 °C.

The fermented almonds were dried in the sun for 7 days until an approximate humidity of 7% was obtained. Afterwards, the grain was taken to a plate roaster, where the almonds were roasted for 45 min. The husk samples of each cocoa variety were obtained with the help of a mechanical huller (10 to 20 min). Both the roasting and husking phase were carried out at the Tropical Experimental Station Pichilingue (Instituto Nacional de Investigaciones Agropecuarias) INIAP. The cocoa fermentation process is summarized in the flowchart (Figure 1).

### 2.3. Sample Preparation

For the analytical determinations, the cocoa husk samples were ground and stored in hermetic containers, isolated from light, moisture and oxygen, for subsequent extraction of bioactive components, physicochemical and functional characterization. Meanwhile, the mucilage samples were thawed at room temperature (18 °C) isolated from light, moisture and oxygen to prevent the oxidation of phenolic compounds and were used directly for their physicochemical and functional characterization.

### 2.4. Determination of the Percentage of Waste

The percentage of residues from each cocoa variety was determined by gravimetry. Ten cacao fruits with known weight were selected. Afterwards, each cob of fruits was opened, the fresh almonds and the placenta were separated and weighed individually. After the fermentation, drying and roasting processes, the nibs and the cocoa husk were weighed. The weight of the mucilage exuded was determined by the difference between the weight of the grain in slime and the grain dried in the sun.

### 2.5. Extraction of Bioactive Compounds in Mucilage and Husk

#### 2.5.1. Cocoa Mucilage

To determine the phenolic components of the cocoa mucilage, the residues exuded from the fermentation of each variety were subject to different extraction processes with and without the help of solvents. It was determined that a centrifugation process at 5200× *g* for 10 min (SIGMA centrifuge, model 4-16Ks, Osterode, Germany) allowed the highest concentration of bioactive compounds to be obtained. The obtained supernatant was used for the quantification of bioactive compounds and antioxidant capacity by the proposed methods. The identification of phenolic compounds and methylxanthines was performed by HPLC analysis. For this, the supernatant obtained was filtered with 0.22 μm Millipore PVDF membranes and stored in vials with a screw cap [28].

#### 2.5.2. Cocoa Bean Shell

For the extraction of the phenolic components of the cocoa bean shell, a 0.3 g sample was weighed and 0.005 L of a solution of methanol, water, formic acid (70:30:0.1%; *v*/*v*/*v*) was added. The extraction process was carried out by four combined cycles of agitation (Mistral Multi-Mixer shaker; Melrose Park, IL, USA) and immersion in an ultrasound bath (Cole-Parmer model 8892; Chicago, IL, USA) for 5 and 10 min, respectively. Then, the sample was centrifuged for 10 min at 2706× *g* at 5 °C (Damon/IEC, Needham Hts, MA, USA). The crude extract from each extraction cycle was collected in a volumetric flask and made up to 25 mL with the extracting solution. The crude extract obtained was used for the quantification of compounds.

### 2.6. Total Polyphenols Content (TPC)

The TPC of exudate mucilage and cocoa bean shell was determined by the method described by Samaniego et al. [5]. For this, 2 mL of each sample was mixed with Folin–Ciocalteu reagent and 20% Na_2_CO_3_. Absorbance was measured using a Shimadzu UV–VIS 2600 spectrophotometer (Kyoto, Japan) at a wavelength of 760 nm. The results were compared against a calibration curve (y = 0.0085x + 0.0975; R^2^ = 0.9989) of gallic acid (0 to 100 ppm). Measurements were made in triplicate for three days (*n =* 9). TPC was expressed in terms of mg of gallic acid (GAE) per 100 mL of sample on a fresh weight (FW) for cocoa exudate mucilage (FD) and dry weight cocoa husk (DW).

### 2.7. Total Flavonoid Content (TFC)

The TFC content of exudate mucilage and cocoa bean shell were determined according to the method described by Llerena et al. [29]. An aliquot of the sample previously extracted in a methanolic solution was taken and reacted with sodium nitrate, aluminum chloride and sodium hydroxide. The absorbance of the mixture was measured using a Shimadzu UV–VIS 2600 Spectrophotometer (Shimadzu, Kyoto, Japan) at 490 nm. TFC was determined using a catechin calibration curve (y = 0.0036x + 0.0678; R^2^ = 0.9994) measured in triplicate for 3 days (*n* = 9). The results are expressed in mg of catechin equivalents (CE) per 100 g dry weight (DW).

### 2.8. Quantification of Phenolic Components Profile of Cocoa By-products by HPLC

The determination of the content of flavan-3-ols was performed by HPLC using a chromatograph (Agilent Technologies 1100/1200 series, Waldbronn, Germany) with a diode array detector (DAD) at a wavelength of 280 nm. The methodology used was previously described by Samaniego et al. [5]; separation was performed using an Agilent Eclipse XDB C18 column (250 mm × 4.6 mm, 5 μm particle size). For the analysis, 20 μL of exudate mucilage and cocoa husk sample were injected into the equipment (automatic injector G1329A) and eluted at a flow rate of 0.8 mL/min using two mobile phases at a temperature of 35 °C. Mobile phase A was composed of acetonitrile/water/formic acid (99/0.8/0.2 *v*/*v*/*v*) and mobile phase B was composed of acetonitrile. For elution with mobile phase A, the analysis was performed in an isocratic mode, while for the mobile phase B, the gradient mode was used (5% to 100% in 67 min). The identification and quantification were carried out by comparison with standards of catechin (CAT), epicatechin (EPI), procyanidin B1 (PCB1), procyanidin B2 (PCB2) and procyanidin C1 (PCC1). The results are expressed in mg of flavan-3-ols per gram of mucilage and husk, respectively.

### 2.9. Quantification of Methylxanthines Theobromine (TBR) and Caffeine (CAF) of Cocoa By-Products by HPLC

The content of TBR and CAF was determined by HPLC (Agilent Technologies series 1100/1200; Waldbronn, Germany), with the help of a DAD detector at a wavelength of 273 nm according to the method described by Samaniego et al. [5], the separation was performed using an Agilent Zorbax SB C18 column (150 mm × 4.6 mm; 5 μm particle size). For the analysis, 20 μL of sample was injected into the equipment (automatic injector G1329A) and eluted at a flow rate of 1 mL/min with a mobile phase composed of a solution of methanol in water at 25% (*v*/*v*). The column oven temperature (G1316A) was 25 °C. The equipment was controlled by Chemstation software (Agilent Technologies, Waldbronn, Germany).

The identification and quantification of the methylxanthines was carried out with TBR and CAF standards. The results are expressed as grams of TBR and CAF per 100 g of mucilage and cocoa bean shell samples on a wet and dry basis, respectively.

### 2.10. Antioxidant Activity of Cocoa By-Products

#### 2.10.1. 2-Azinobis (3-Ethyl-benzothiazoline-6-sulfonic acid) Cation Bleaching (ABTS) Assay

The antioxidant activity of exudate mucilage and cocoa bean shell (CCN-51 and Nacional X Trinitario type) was evaluated using the ABTS method described by Llerena et al. [29]. A stock solution was prepared, composed of a potassium persulfate solution (2.45 mM) and an ABTS solution (7 mM) in a ratio (1:1, *v*/*v*), the mixture was incubated for 16 h at room temperature in the dark. From the stock solution, a working solution was prepared in a ratio (1:25, *v*/*v*) with a phosphate buffer solution at pH 7.0 (75 mmol/L monobasic sodium phosphate 0.2 mol/L and dibasic sodium phosphate 0.02 mol/L), until an absorbance reading of 1.1 ± 0.01 at a wavelength of 734 nm was obtained.

Then, 200 µL of crude extract (centrifuged cocoa mucilage) was mixed with 3800 µL of the working solution. With the help of a stirrer (Vortex Multimixer Lab-Line Instruments), the mixture was homogenized and incubated for 45 min at room temperature and in the dark.

The antioxidant activity was quantified by UV–VIS spectrophotometry (Shimadzu 2200 Spectrophotometer; Tokyo, Japan) against a Trolox standard curve (0–800 µmol/L). The curve obtained was y = 0.0013x + 0.126; R^2^ = 0.9982. All assays were performed in triplicate (*n* = 9) for 3 days. The results are expressed as µM of Trolox equivalents (TE)/mL of sample for mucilage sample and mg TE/g DW for bean shell sample. 

#### 2.10.2. Ferric-Reducing Antioxidant Power (FRAP) Assay

The antioxidant activity of exudate mucilage and cocoa bean shell samples was evaluated by the FRAP method. The tested methodology was described by Samaniego et al. [30]. For the determination of the antioxidant activity, 1 mL of each raw extract of previously diluted cocoa mucilage and husk was taken. These samples were mixed with 2.5 mL of phosphate buffer solution pH 6.6 (monobasic sodium phosphate solution 2.4% (*w*/*v*) and dibasic sodium phosphate 2.84% (*w*/*v*), in distilled deionized water). A measure of 2.5 mL of 1% (*w*/*v*) potassium ferrocyanide solution was added to the mixture and it was incubated at 50°C in a water bath (Memmert, Schwabach; Germany) for 20 min. After incubation, 2.5 mL of 10% (*w*/*v*) trichloroacetic acid solution, 2.5 mL of distilled water and 0.5 mL of 1% (*w*/*v*) ferric chloride were added. The solution obtained was left to stand for 30 min at room temperature and in the dark.

The antioxidant activity was quantified by UV–VIS spectrophotometry (Shimadzu 2200 Spectrophotometer; Tokyo, Japan) at 700 nm against a Trolox standard curve (0–800 µmol/L). The calibration curve was established at (y = 0.0016x + 0.1018, R^2^ = 0.998). Results are expressed as µM TE/mL of sample for mucilage sample mg TE/g DW for bean shell sample. 

#### 2.10.3. Oxygen Radical Absorbance Capacity (ORAC) Assay

The determination of the antioxidant activity by the ORAC method was carried out following the method reported by Piñuel et al. [31]. Analytical measurements were performed using Biotek Sinergytm HT microplate spectrofluorometer (BMG Labtech, Offenburg, Germany) with fluorescence filters for excitation (485 nm) and emission (520 nm), using 96-well flat-bottom black microplates (Brand Plates, Werthein, Germany). A measure of 25 µL of both Trolox standards and exudate mucilage and cocoa bean shell samples were placed on the plates. A measure of 25 µL of buffer pH 7.43 was added. Subsequently, the plate was placed inside the support of the spectrofluorometer and the analysis began. Then, 150 μL of fluorescein (17.7 nM) was added, and the solutions were incubated at 37 °C for 30 min. After, 25 μL of the AAPH solution (125.92 mM) was added. Before measuring fluorescein intensity, the plate was shaken (15 s) and fluorescence measurements began to be recorded every minute for 45 min. For the determinations, the area under the curve (AUC) of the standards and samples was calculated using the Gen5 Data Analysis software. ORAC values were calculated using a regression equation between Trolox concentration and AUC (y = 0.4174x + 15.795; R^2^ = 0.9976). Values are expressed in 0–700 μM TE/mL for mucilage sample and mg TE/g for bean sample.

### 2.11. Statistical Analysis

To evaluate the effect of the cocoa variety on the content of total antioxidants (polyphenols and flavonoids), antioxidant activity, phenolic profiles (procyanidin B1, procyanidin B2, procyanidin C1, epicatechin, catechin) and methylxanthines of mucilage and cocoa bean shells of the varieties CCN-51 and National X Trinitario type, the Student’s *t*-test was applied. The determinations were made with a confidence level of 95% (∞ = 0.05). In addition, a Pearson correlation analysis was performed using the variables in two sets, one corresponding to the concentrations of 3-flavan-ols and methylxanthines and a second group with the values of antioxidant activity evaluated by the ABTS, FRAP methods. and ORAC. All values are expressed as the mean ± standard deviation (SD). Assays were carried out in triplicate for three days (*n* = 9). Statistical differences are indicated with superscript lowercase letters.

## 3. Results and Discussion

### 3.1. Percentage of Waste from the Cocoa Processing Process

Each cocoa fruit of the CCN51 variety was previously weighed. CCN-51 cocoa pods present an average weight of 814 g, 27 g of placenta, 226 g of fresh almonds. Each cob presented approximately 49 grains. Figure 2 shows the percentages of cocoa residues in the different parts of the fruit. The CCN-51 cocoa has higher percentages of residues of cocoa beans, mucilage, bean shells and cocoa placenta than the National X Trinitarian type cocoa. For the CCN51 and Nacional X Trinitario-type cocoa varieties, the pod husk is the main residue of the cocoa fruit at 68.40 and 79.90%, respectively. The second residue of the cocoa fruits is the cocoa bean with percentages of 20.90 and 14.45%, for CCN51 and Nacional X Trinitario type. Cocoa residue percentages have been reported with values for cocoa pod shells representing 67–76% of the whole fruit, and percentages for cocoa beans representing about 33% of the whole fruit [32,33]. The value of pod husks of the Nacional X Trinitario-type variety was slightly higher than those reported in the literature, but the values for the cocoa beans for the two varieties studied were significantly lower than those reported in the literature. These differences may be due to the varieties studied and the differences in the waste separation processes. Figueroa et al. [34] and Campos-Vega et al. [35] reported cocoa fruit composition values with proportions of 67–76% for pod husks, 30–40% for cocoa beans, 8.7–9.9% for cocoa mucilage and 2.1–2.3% for coca bean shells. The percentages reported in this study for the varieties CCN-51 and Nacional X Trinitario type of pod husks, cocoa beans, cocoa mucilage and cocoa bean shells were lower than those previously described.

### 3.2. Total Polyphenol Content (TPC) and Total Flavonoid Content (TFC) Composition of Cocoa By-Products

Table 1 shows the content of TPC and TFC in the mucilage and cocoa bean shells of the cocoa varieties CCN-51 and Nacional X Trinitario type. The TPC and TFC values in the mucilage samples of the Nacional X Trinitario-type cocoa variety were higher than the values of the CCN-51 cocoa variety. The cocoa bean shell samples of the CCN-51 cocoa variety presented the highest values of TPC and TFC.

The results obtained in this study were in the range reported by Guirlanda et al. [36] and Martinez et al. [37], who reported values in the range of 101.50 to 182.63 mg GAE/100 g DW for cocoa samples from the Nacional X Trinitario complex from Brazil (Buerarema, Ilhéus Bahia) and Ecuador (Cono and Taura in the province of Guayas), respectively. Regarding by-products of the CCN-51 cocoa variety, no specific values were found for the content of TPC or for TFC.

According to the values reported in Table 1, there are statistically significant differences (*p* < 0.05) between the two cocoa varieties in their mucilage and cocoa bean shell samples. Samaniego et al. [5] proposes that there is an effect of the genotype, the harvest season, the place of origin and the postharvest processes on the antioxidant content of cocoa beans. Considering that these secondary metabolites accumulate in different parts of the plant, the behavior of the almonds is reflected in the cocoa residues. For example, Hernández-Hernández et al. [38], described that the TPC varied in the cotyledons of 25 genotypes from the INIFAP germplasm bank in Mexico between 417.02 and 1157.35 mg GAE/100 g. In samples from crosses with Ecuadorian cocoa INI11 (RIM76A × EET400), INI10 (POUND7 × EET400) and INI18 (RIM76A × EET48), these values can vary between 532 and 1117 mg GAE/100 g DW, being higher than those found in this study. This may be because the TPC decreases from 100 to 10% during fermentation [39].

The content of TPC can be affected by other factors such as the extraction method (solvents, sonication time, extraction temperature). According to the work of Martínez et al. [37], the samples of methanol-acetone extracts (173.67 to 182.63 mg GAE/100 g DW) presented a higher recovery of metabolites than in ethanolic medium (102.00 to 109.00 mg GAE/100 g DW) in Nacional X Trinitario-type cocoa mucilage, from Cono and Taura in the province of Guayas, Ecuador. Nsor-Atindana et al. [40] described TPC and TFC content in cocoa bean shell extracts. The extracts were prepared with different solvents, such as acetone, ethanol, and methanol. TPC values reported were in a range of 1715–4182 mg GAE/100g DW.

For TFC content, they presented a behavior similar to TPC, with a higher concentration in mucilage exuded from cocoa type Nacional X Trinitario (37.00 ± 0.08 mg CE/100 g) than in mucilage of the CCN-51 variety (5.00 ± 0.03). These values were higher than those reported by Guirlanda et al. [36], who presented a flavonoid content of 0.72 mg/100 mL for cocoa samples from the Nacional X Trinitario complex from Brazil (Buerarema, Ilhéus Bahia). Lessa et al. [41] presented a TFC content within the range presented in this study for cocoa mucilage, using hydroalcoholic extraction (23 mg CE/100 mL) and aqueous (21 mg CE/100 mL) media. These authors considered that there was an effect from the extraction method on the phytocompounds. It was clear that the type of solvent used affected the concentration of the phytocompounds.

The cocoa bean shell samples presented a different behavior from the cocoa mucilage samples, where the CCN-51 variety presented a higher content of total antioxidants than the samples from the Nacional X Trinitario-type variety for both TPC (42.17 and 29.43 mg GAE/100 g) as well as for TFC (20.57 and 15.08 mg CAT/100 g), finding statistically significant differences (*p* < 0.05) due to the effect of the variety of origin of the residues with a confidence level of 95.0%.

The TPC found in this study is higher than the values obtained by Martínez et al. [37], who presented values between 144.48 and 154.43 mg GAE/100 g for samples of Nacional X Trinitario de Taura and Cono cocoa, respectively. However, these values are within the range reported by other authors [13,41] 611 to 5780 mg GAE/100 g for cocoa from Ecuador and Brazil. According to these authors, the cocoa benefit chain residues show much lower values than the almonds before and after fermentation. The presence of these compounds in the residues is due to the fact that the polyphenols can migrate from the cotyledons to the hull, generating the enrichment of these by-products [39]. The differences found in the content of TPC in the samples of cocoa hulls evaluated in this study could be related to the variety, harvest season and roasting, which can affect the loss of these compounds at different levels [42].

Lessa et al. [41] reported a TFC content of 0.0015 mg quercetin equivalents (QE)/100 g; these values were below the flavonoid content found in this study. However, these authors mention that both almonds and residues had high levels of flavonoids, before and after processing. Md Yusof et al. [43] described the TFC content from cocoa shell extracts with a value of 7.47 mg rutin equivalents (RE)/g DW. These extracts were obtained via an ultrasound-assisted extraction method using ethanol as solvent. The TFC values of cocoa by-products in this study were higher than in the work previously described. Barbosa-Pereira et al. [44] obtained cocoa bean shell extracts using the technique known as pulsed electric field-assisted extraction. They found TPC values between 24.93 and 32.30 mg GAE/g DW; these values were higher than our values. These pronounced differences may be due to differences in the techniques used to obtain the extracts. Felice et al. [45] reported 7105 mg GAE/100 g FW in cocoa husk extracts. Razola-Diaz et al. [46] indicated the TPC of cocoa from different countries. For example, cocoa beans from Peru had a TPC of 57.40 mg GAE/g DW; cocoa beans from Venezuela showed a TPC of 34.4 mg GAE/g DW while West African cocoa beans had a TPC of 9.2 mg GAE/g DW. Our TPC in cocoa by-products was higher than those reported in these extracts. When compared with the values of cocoa beans from Colombia (44–202 mg GAE/g DW), we are below these values even though this study is on whole beans and not on by-products [47,48,49]. The same occurs when we compare our values with the TPC of cocoa beans from Ecuador (33.6–71.7 mg GAE/g DW) [5].

### 3.3. Profile of Phenolic Components 

Figure 2 shows the phenolic compounds identified in cocoa by-products using the HPLC technique. The phenolic compounds were identified by the retention time of their respective standards. The first component to elute in the chromatogram of the standards was PCB1 (17.53 min), followed by CAT (19.83 min), PCB2 (21.22 min), EPI (23.11 min), and PCC1 (24.35 min) (Appendix A). The mucilage samples of the varieties CCN-51 and Nacional X Trinitario type presented the same phenolic profile with small changes in retention times. Compounds such as PCB1 (16.20 min), CAT (18.88 min), PCB2 (20.41 min) were identified in CCN-51 cocoa exudate mucilage (Figure 3a); EPI (22.88 min) and PCC1 (23.57 min). For the National cocoa exudate mucilage (Figure 3b), PCB1 appears at 16.18 min, followed by the peaks of CAT (18.88 min), PCB2 (20.36 min); EPI (22.2 min) and PCC1 (23.53 min).

In the cocoa bean shells of the two varieties studied, only CAT and EPI were identified (Figure 3c,d). The peaks with the highest intensity corresponded to the CAT content in the two varieties. When comparing the results obtained in this research with the work carried out by Samaniego et al. [5], it can be seen that the mucilage samples had the same phenolic profile as the cocoa beans studied by these authors. Manuela et al. [16] described the profile of the phenolic components of cocoa by-products (cocoa beans) with the presence of PCB1, PCB2, CAT, EPI and protocatechuic acid. The study did not indicate the variety of cocoa or its origin. The profile of the polyphenolic compounds of the cocoa by-products (cocoa bean shells) was reported with HPLC and MS analysis. In these reports, it was described that EPI was the most abundant compound in this by-product with a considerable concentration of CAT, followed by its dimers, PCB1 and PCB2. [17,50,51].

### 3.4. Profile of Methylxanthines

Figure 3 shows the profile of methylxanthines present in the extracts of different cocoa residues. The methylxanthines in the cocoa by-products were identified with the help of the retention times of the TBR and CAF standards (Appendix A). TBR and CAF were identified in the mucilage samples from the cacao varieties CCN-51 and cocoa by-products of the Nacional X Trinitario type. The retention times for TBR and CAF in the mucilage samples of the two varieties were very similar (Figure 4a,b). The Nacional X Trinitario-type variety presented a greater intensity for both compounds.

In the cocoa bean shells, TBR and CAF were also identified (Figure 4c,d). The retention times of TBR and CAF for both varieties were the same. The peaks with the highest intensity in both varieties of cocoa bean shells corresponded to TBR.

### 3.5. Quantification of Phenolic Components

Table 2 shows the content of phenolic compounds in cocoa by-products. The mucilage sample from the National X Trinitario-type variety presented the majority values of PCB1, PCB2, PCC1, PPE and CAT. The highest values corresponded to PCB2 (3.52 mg/100 mL) and CAT (3.54 mg/100 mL). When comparing the concentrations of EPI and CAT between the cocoa mucilage and cocoa bean shell samples, it was observed that the highest values corresponded to the cocoa bean shell sample of the two varieties studied. The highest value corresponded to the CAT of the cocoa bean shell sample of the National × Trinitario-type variety with a CAT value of 1616.2 mg/100 mL. The statistical analysis showed no significant differences when comparing the concentrations of all the polyphenols identified in the cocoa mucilage sample of both varieties studied. The concentrations of EPI and CAT in the cocoa bean shell samples of the two varieties did show significant differences when compared with each other. There are significant differences (*p* < 0.05) when comparing the same sample of different varieties. Rossin et al. [50] described the quantification of polyphenolic compounds from by-products (cocoa bean shell). They indicated that EPI (0.21–34.97 mg/g) was the most abundant compound followed by CAT (0.18–4.50 mg/g). In our study, the CAT and EPI of the cocoa bean shells of the two cocoa varieties presented higher values than those reported by Rossin et al. [50].

According to Martinez et al. [37] and Okiyama et al. [39], the most typical and abundant phenolic compounds in cocoa and its residues are flavan-3-ols such as CAT, EPI and procyanidins. These compounds are responsible for the bitter taste of almonds, cocoa derivatives and residues. According to the study by Hernandez-Hernandez et al. [38], cocoa cotyledon is rich in CAT (148 mg GAE/100 g) and EPI (613 mg GAE/100 g). For Okiyama et al. [39], the cocoa bean shell is rich in phenolic compounds (soluble polyphenols and tannins). This author mentions that cocoa residues are rich in proanthocyanidins and tannins that can vary in the size of their monomers; however, these are not found in the cocoa bean shell, confirming the results found in this study.

Unlike cocoa exudate mucilage, the cocoa bean shell does not have the same phenolic profile as cocoa beans, presenting phenolic compounds such as CAT and EPI in both varieties. In the Nacional X Trinitario-type cocoa, the shell presented concentrations of 16.26 mg/g of CAT and 0.83 mg/g of EPI. CAT (4.99 mg/g) and EPI (0.84 mg/g) values were obtained in cocoa bean shells from the CCN-51 variety.

### 3.6. Methylxanthines Content

Table 3 shows the content of methylxanthines in cocoa by-products. Methylxanthines were found in higher concentration in cocoa mucilage and cocoa bean shells, the highest of which was TBR. This was observed in the two varieties studied. The highest value was for TBR (2.66 mg/100 mL) and corresponds to the cocoa mucilage sample of the Nacional X Trinitario-type variety. Statistical analysis showed significant differences when comparing CAF and TBR concentrations. It also reflected statistical differences when comparing the same compound in the two varieties.

Due to the fact that there is very little information regarding the study of methylxanthines in cocoa mucilage and the bean shell, reference data from cocoa [5] were taken, where the alkaloid content of cocoa beans ranged between 1.52 and 1.85 (g/100 g) in TBR and CAF, respectively. These methylxanthines have been used as a good marker to classify cocoa beans by their genotype, also constituting an important tool to certify the quality of the beans. There is no record regarding the study of the content of alkaloids in cocoa mucilage. However, authors such as Okiyama et al. [39] and Lessa et al. [29] showed that cocoa bean shell extracts are rich in alkaloids or methylxanthines such as theobromine, caffeine and theophylline. According to Vasquez et al. [13], this cocoa processing chain residue contains 1.2 g/100 g TBR and 0.12 g/100 g CAF, presenting lower values (TBR 2.4–3.2 g/100 g and CAF 0.3 to 1 g/100 g) than those obtained by other authors for fresh cocoa beans [13,52].

### 3.7. Antioxidant Activity 

The antioxidant activity of cocoa benefit chain byproducts depends on phytochemical compounds such as polyphenols, and flavan-3-ols such as catechin, epicatechin, and procyanidins. The antioxidant activity of cocoa benefit chain residues (mucilage and cocoa bean shell) measured by three ABTS, FRAP and ORAC methods, is shown in Table 4.

According to the results obtained, the cocoa mucilage from the CCN-51 variety presented lower values of antioxidant activity (4.69, 3.35, and 1.28 µM TE/mL) than the National X Trinitario-type cocoa samples (8.54, 7.89, and 1.33 µM TE/mL). In contrast, in the cocoa bean shells, the antioxidant activity values measured by the three methods were higher in samples from the CCN-51 variety than in samples from the Nacional X Trinitario-type cocoa. In the two by-products (cocoa mucilage and shells), the results showed that there are statistically significant differences (*p* < 0.05) between the two varieties, evidencing that the residues’ variety of origin affects the antioxidant activity (*p* < 0.05), for all methods.

According to Martinez et al. [37], the antioxidant activity of cocoa by-products (pod shell, kernel and mucilage) could be attributed to the presence of bioactive compounds such as CAT, EPI and PCB2. These authors reported antioxidant activity values of 4.10 to 4.17 µM TE/mL in ABTS and for FRAP for cacao mucilage of the Nacional X Trinitario type from the Yaguachi and Naranjal cantons, Guayas Province (Pacific Coast). The difference in the results obtained in this research compared to the work presented by Martínez can be attributed to several factors such as the type of solvents used in the extraction process and the geographical area of origin of the sample. Luna-Guevara et al. [53], and Mazzutti et al. [54] found that the results of the antioxidant activity tests may be conditioned by the extraction methods (methanol/acetone and acetone/water/acetic acid) used, since, in most cases, 100% of the analytes responsible for the antioxidant activity are not recovered. At the bibliographic level, there is very little information regarding the antioxidant activity of cocoa mucilage measured by the ORAC method. Razola-Diaz et al. [6] described the antioxidant activity of cocoa beans of different origins. They found DPPH values ranged from 30.77 to 97.94 mg TE/g DW. For ABTS, the values were from 73.97 to 267.43 mg TE/g D.W., and for the FRAP assay from 28.88 to 98.74 mg TE/g DW. When comparing the antioxidant activity of the bean shell samples by the ABTS and FRAP methods with the previous study, it could be observed that the said activity was higher.

### 3.8. Pearson Correlation Analysis

Table 5 presents the results of the Pearson correlation analysis (r) between the antioxidant activity measured by ABTS, FRAP and ORAC with the contents of methylxanthines (TBR and CAF) and the bioactive compounds that are part of the phenolic profile identified in the exuded mucilage (PCB1, PCB2, PCC1, EPI and CAT) and cocoa bean shell (EPI and CAT). The antioxidant activity of cocoa mucilage measured by the ABTS method showed a positive and statistically significant correlation (*p* < 0.05) with the content of phenolic compounds and methylxanthines, reporting a strong positive correlation (r = 0.95 to 0.99 or 0.95 to 0.99) with PCN-B1 (0.9993), PCB2 (0.9982) and PCC1 (0.9997), EPI (0.9985) and CAT (0.99, 0.95). Similarly, the antioxidant capacity of cocoa mucilage measured by the FRAP method presented a strong positive correlation with the aforementioned bioactive compounds (r ≥ 0.99). In the case of PCC1, it had a less strong correlation (0.9702) but no less important. With the ORAC method, the Pearson correlation percentages were lower than those obtained by the ABTS and FRAP methods, obtaining values of 0.9501, 0.9489, 0.9449, 0.9493 and 0.9220 for PCB1, PCB2, EPI, CAT and PCC1, in that order.

In cocoa bean shells, a different behavior was observed than in cocoa mucilage. In cocoa bean shells, it was determined that the antioxidant capacity measured by ABTS and FRAP presented a strong positive correlation (r = 0.95–0.99 or 0.95–0.99) with the catechin content with values of 0.9675 and 0.9825, respectively. However, the antioxidant capacity measured by the ORAC method presented a moderate positive correlation (r = 0.50–0.94 or 0.50–0.94) with this phenolic component (r = 0.8747). When correlating the EPI with the antioxidant activity, it was observed that there was no correlation (r = 0.00–0.09 or 0.00–0.9) between this phenolic component and the antioxidant capacity measured by the ABTS (1.30) and FRAP (0.574) methods. While the correlation of these components with the antioxidant capacity measured by ORAC (−0.2595) was weak and negative (r = 0.10–0.49 or 0.10–0.49).

According to the results of the Pearson correlation analysis, the antioxidant activity of the biochemical compounds found in the cocoa by-products could contribute as antioxidant agents, whose biological function is to trap free radicals present in our body (ABTS). In addition, they have the ability to reduce the metallic compounds that act as pro-oxidants in our body by favoring the action of antioxidant compounds (FRAP). Similarly, they capture peroxyl radicals, one of the reactive oxygen species (ROS) responsible for protein damage, DNA, RNA, among others (ORAC) [25,26,27].

Although the antioxidants identified in cocoa bean shells were CAT and EPI, only catechin showed significant correlations with the antioxidant activity. Therefore, in vitro, the bioactive properties of cocoa bean shells showed that catechin is responsible for chelating redox active metal ions, inactivating lipid free radical chains, suppressing the conversion of hydroperoxide to reactive oxyradicals, causing the disappearance of chromogenic radicals.

TBR and CAF are generally studied as alkaloids of natural origin, with a stimulating effect on the central nervous system, physiological effects and cardiovascular benefits such as the reduction of blood pressure and LDL cholesterol levels in the blood. However, it has been shown that these compounds can act effectively under conditions of oxidative stress, reducing the anti-inflammatory effect by suppressing pro-inflammatory cytokines. Both TBR and CAF are highly related to the in vitro antioxidant activity measured by the three methods in exudate cocoa mucilage samples: ABTS (>0.99), FRAP (>0.99) and ORAC (>0.94) [36,37,38,42]. Similarly, in the cocoa bean shell samples, a different behavior was observed from that of the cocoa exudate mucilage, presenting a strong positive correlation (r = 0.95–0.99 or 0.95–0.99) between the contents of CAF and TBR with the antioxidant capacity measured by the methods of ABTS (0.9867, 0.9722) and FRAP (0.9828–0.9833) and a moderate positive correlation (r = 0.50–0.94 or 0.50–0.94) with the antioxidant capacity measured by ORAC (0.8629–0.8730).

## 4. Conclusions

In this study, it was possible to establish that bioactive cocoa compounds are present in the external parts of cocoa fruits (bean shells, pods husks, mucilage and placenta) which are considered cocoa by-products.

HPLC analysis allowed us to identify and quantify the profile of the phytonutrients in cocoa mucilage and bean shells (CAT, PCB1, PCB2, PCC1 and EPI) and methylxanthines (TBR and CAF) in the varieties CCN-51 and Nacional X Trinitario type grown in Ecuador. The mucilage and bean shell residues of the two cocoa varieties showed a strong antioxidant activity. Pearson’s correlation analysis allowed us to establish a positive relationship between the content of polyphenols, flavonoids and antioxidant activity. This biological activity may add value to cocoa residues and allow them to be considered as an important source of phenolic components. Cocoa residues could be used as functional ingredients for different purposes in the food industry.

## Figures and Tables

**Figure 1 foods-12-02583-f001:**
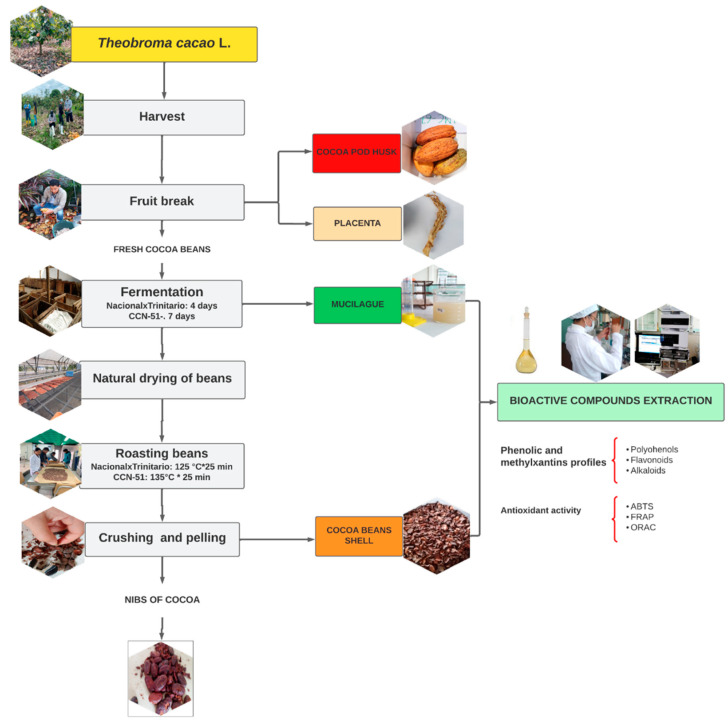
Flowchart of cocoa fermentation process.

**Figure 2 foods-12-02583-f002:**
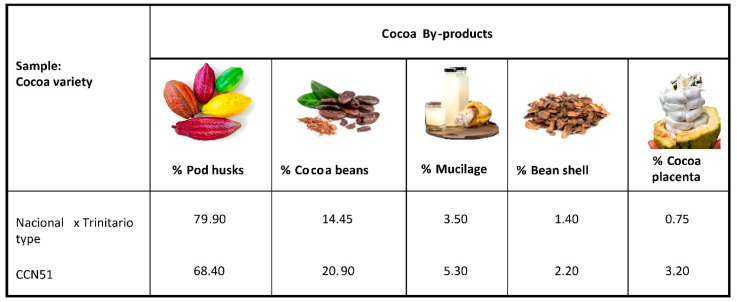
Percentage of cocoa by-products of CCN-51 and Nacional X Trinitario-type cocoa varieties.

**Figure 3 foods-12-02583-f003:**
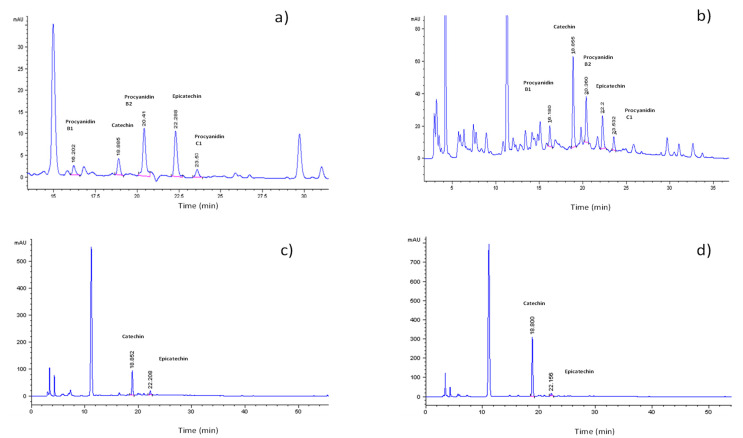
Profile of phenolic components of cocoa by-products analyzed by HPLC. (**a**) Cocoa mucilage of CCN-51 variety; (**b**) cocoa mucilage of Nacional X Trinitario-type variety; (**c**) cocoa bean shell of CCN-51 variety; (**d**) cocoa bean shell of Nacional X Trinitario-type variety.

**Figure 4 foods-12-02583-f004:**
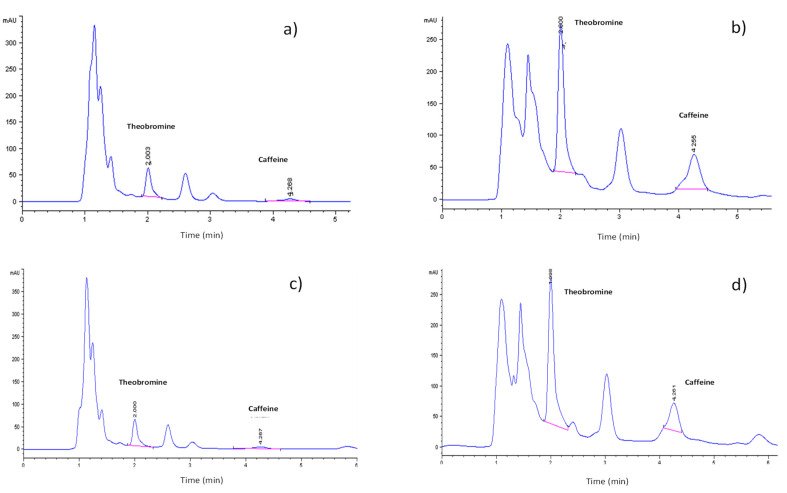
Profile of methylxanthines of cocoa by-products analyzed by HPLC. (**a**) Cocoa mucilage of CCN-51 variety; (**b**) cocoa mucilage of Nacional X Trinitario-type variety; (**c**) cocoa bean shell of CCN-51 variety; (**d**) cocoa bean shell of Nacional X Trinitario-type variety.

**Table 1 foods-12-02583-t001:** Quantification of total polyphenol content (TPC) and total flavonoid content (TFC) on cocoa by-products.

Phenol Component	Variety	Cocoa Mucilage	Cocoa Bean Shell *
Total polyphenol content (TPC) (mg GAE/100 mL)	CCN-51	72.22 ± 1.53 ^a^	42.17 ± 16.10 ^a^
	Nacional X Trinitario type	105.08 ± 3.54 ^b^	29.43 ± 66.44 ^b^
Total flavonoid content (TFC) (mg CE/100 mL)	CCN-51	5.16 ± 0.03 ^a^	20.57 ± 15.30 ^a^
	Nacional X Trinitario type	36.80 ± 0.08 ^b^	15.08 ± 33.09 ^b^

Results are expressed as mean ± standard deviation (SD) (*n* = 9) and were analyzed by Student’s *t*-test. Statistical differences (*p* < 0.05) are indicated with superscript letters. GAE: gallic acid equivalents; CE: catechin equivalents. * Cocoa bean shell is expressed in mg GAE/100 g DW.

**Table 2 foods-12-02583-t002:** Quantification of phenolic components of cocoa by-products (mucilage and bean shell).

Phenolic Components	Cocoa Mucilage (mg/100 mL)		Cocoa Bean Shell (mg/100 g)	
	CCN-51	Nacional X Trinitario Type	CCN-51	Nacional X Trinitario Type
Procyanidin B1 (PCB1)	0.55 ± 0.001 ^a^	2.56 ± 0.05 ^b^	N.D	N.D
Procyanidin B2 (PCB2)	0.92 ± 0.001 ^a^	3.51 ± 0.13 ^b^	N.D	N.D
Procyanidin C1 (PCC1)	0.60 ± 0.001 ^a^	1.68 ± 0.24 ^b^	N.D	N.D
Epicatechin (EPI)	0.66 ± 0.01 ^a^	1.37 ± 0.02 ^b^	84.25 ± 1.51 ^a^	83.90 ± 2.11 ^a^
Catechin (CAT)	0.29 ± 0.003 ^a^	3.54 ± 0.04 ^b^	499.45 ± 0.02 ^a^	1616.62 ± 0.01 ^b^

Results are expressed as mean ± standard deviation (SD) (*n* = 3) and were analyzed by Student’s *t*-test. Statistical differences (*p* < 0.05) are indicated with superscript letters. N.D: not determined.

**Table 3 foods-12-02583-t003:** Content of methylxanthines of cocoa by-products (mucilage and bean shell).

Methylxanthines	Cocoa Mucilage (g/100 mL)		Cocoa Bean Shell (g/100 g)	
	CCN-51	Nacional X Trinitario Type	CCN-51	Nacional X Trinitario Type
Caffeine	0.12 ± 0.002 ^a,A^	0.91 ± 0.041 ^a,B^	0.11 ± 0.002 ^a,A^	0.36 ± 0.001 ^a,B^
Theobromine	0.49 ± 0.013 ^b,A^	2.66 ± 0.089 ^b,B^	0.93 ± 0.013 ^b,A^	1.29 ± 0.022 ^b,B^

Results are expressed as mean ± standard deviation (SD) (*n* = 3) and were analyzed by Student’s *t*-test. Statistical differences (*p* < 0.05) are indicated with superscript letters.

**Table 4 foods-12-02583-t004:** Antioxidant activity of cocoa by-products of cocoa varieties CCN-51 and Nacional X Trinitario type by ABTS, FRAP and ORAC methods.

By-Product	Cocoa Variety	Antioxidant Activity		
		ABTS	FRAP	ORAC
Mucilage (µM TE/mL)	CCN-51	4.69 ± 0.01 ^a^	3.35 ± 0.06 ^a^	1.28 ± 0.02 ^a^
	Nacional X Trinitario type	8.54 ± 0.00 ^b^	7.89 ± 0.10 ^b^	1.33 ± 0.00 ^a^
Bean shell (mg TE/g)	CCN-51	171.32 ± 3.09 ^a^	192.22 ± 3.20 ^a^	56.87 ± 1.62 ^a^
	Nacional X Trinitario type	167.06 ± 1.50 ^b^	160.06 ± 3.87 ^b^	52.53 ± 1.34 ^b^

Results are expressed as mean ± standard deviation (SD) (*n* = 9) and were analyzed by Student’s *t*-test. Statistical differences (*p* < 0.05) are indicated with superscript letters. TE: Trolox equivalents.

**Table 5 foods-12-02583-t005:** Pearson’s correlation analysis between phytocompounds and antioxidant activity of cocoa by-products.

Phytocomponents	Cocoa Mucilage			Cocoa Bean Shell		
	ABTS	FRAP	ORAC	ABTS	FRAP	ORAC
Procyanidin B1	0.9993	0.9999	0.9501	N.D	N.D	N.D
Procyanidin B2	0.9982	0.9967	0.9489	N.D	N.D	N.D
Procyanidin C1	0.9697	0.9702	0.922	N.D	N.D	N.D
Epicatechin	0.9985	0.9981	0.9449	1.30	0.574	−0.2595
Catechin	0.9995	0.9991	0.9493	0.9675	0.9825	0.8747
Caffeine	0.9979	0.9994	0.9474	0.9867	0.9828	0.8629
Theobromine	0.9987	0.9973	0.9498	0.9722	0.9833	0.8730

N.D: not determined.

## Data Availability

Not applicable.

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
