# Peer review of "Profile of Bioactive Components of Cocoa (Theobroma cacao L.) By-Products from Ecuador and Evaluation of Their Antioxidant Activity"

_foods, 2023, doi:10.3390/foods12132583_

Round 1

Reviewer 1 Report

This article focuses on Bioactive components of Cocoa By-products from Ecuador and Evaluation of their Antioxidant Activity. The article is interesting and meets the criteria of scientific novelty. The volume and subject of the article meets the requirements of the Journal. The prospects and relevance of the topic are due to the possibilities of practical application of this knowledge and their possible further development. Despite the advantages of this work, there are some points that can be improved:

1. Abstract can be expanded.

2. It is desirable to justify the choice of research objects in more detail.

3. What substances (included in the studied plants) make the maximum contribution to the antioxidant activity?

4. More comparison of the obtained data with literary sources is desirable.

Author Response

Reviewer # 1:

Comments and Suggestions for Authors

This article focuses on Bioactive components of Cocoa By-products from Ecuador and Evaluation of their Antioxidant Activity. The article is interesting and meets the criteria of scientific novelty. The volume and subject of the article meets the requirements of the Journal. The prospects and relevance of the topic are due to the possibilities of practical application of this knowledge and their possible further development. Despite the advantages of this work, there are some points that can be improved:

  1. Abstract can be expanded.

Answer: The Abstract was expanded. The antioxidant values were added. “Abstract: The aim of the study was to determine the profile of bioactive compounds in cocoa residues (mucilage and bean shell), and to evaluate their antioxidant activity in two cocoa varieties, Nacional x Trinitario type (Fine Aroma) and the variety CCN-51. The extraction of phytonutrients from the residues was carried out selectively. The characterization and quantification of the total polyphenols content (TPC), and the total flavonoids content (TFC) were determined by UV-VIS spectrophotometry. High-Performance Liquid chromatography (HPLC) was used to determine the phenolic profile and methylxanthines.

The antioxidant activity was evaluated by the 2-azinobis (3-ethyl-benzothiazoline-6-sulfonic acid) cation bleaching (ABTS), ferric-reducing antioxidant power (FRAP) and oxygen radical absorbance capacity (ORAC) methods. The exudate mucilage samples from Nacional x Trinitario type cocoa presented the highest content of TPC 105.08 mg gallic acid equivalents (GAE)/100 mL, TFC 36.80 mg catechin equivalents (CE)/100 mL, catechin (CAT) 35.44 mg/g; procyanidins (PCB2: 35.10; PCB1: 25.68; PCC1: 16.83 mg/L), epicatechin (EPI) 13.71 mg/L, caffeine (CAF) 0.90% and theobromine (TBR) 2.65%. In the cocoa bean shell, the variety CCN-51 presented a higher content of TPC (42.17 mg GAE/100g) and TFC (20.57 mg CE/100g). However, CAT (16.16 mg/g), CAF (0.35%) and TBR (1.28%) were higher in the Nacional x Trinitario cocoa type. The EPI presented no significant differences between the two samples studied (0.83 and 0.84 mg/g). The antioxidant activity values (ABTS, FRAP and ORAC methods) were higher in the samples of CCN-51 than Nacional x Trinitario type. The bean shell samples presented antioxidant values of (171.32; 192.22 and 56.87 mg trolox equivalents (TE)/g) respectively and the bean shell samples presented antioxidant values of (167.06; 160.06 and 52.53 mg TE/g) respectively. The antioxidant activity (ABTS, FRAP and ORAC) of the residues was correlated with the bioactive compounds of mucilage and bean shell, showing a strong positive correlation (<0.99) with the procyanidins (B1, B2 and C1), EPI and CAT and positive/moderate correlation (0.94) with methylxanthines”.

  1. It is desirable to justify the choice of research objects in more detail.
    Answer: the new justify was added
  2. What substances (included in the studied plants) make the maximum contribution to the antioxidant activity?

In this study, the antioxidant activity of the cocoa extracts of the two varieties was evaluated. These extracts are rich in polyphenolic compounds and flavonoids. The antioxidant activity of phenolic compounds is widely described in the literature. “Antioxidant activity of cocoa beans is contributed mainly by polyphenols (Pedan, Fischer, and Rohn 2016) and Maillard reaction products (Ioannone et al. 2015). Pedan, Fischer, and Rohn (2016) found that methylxanthines (such as theobromine and caffeine) had no reactions with DPPH radicals. Theobromine and caffeine showed antioxidant activity in calf thymus DNA (in vitro) by inhibiting oxidative DNA breakage and the generation of hydroxyl radicals (Azam et al. 2003). Thus, the contribution of alkaloids to the antioxidant activity was dependent on the types of assays. (Febrianto, Wang & Zhu, 2021)”. Mainly the antioxidant activity of cocoa by-products is due to phenolic compounds. “Baranowska et al. (2020) showed that AA of cocoa ethanolic extract was due to synergistic effects of catechin, epicatechin, gallocatechin, procyanidin B1, procyanidin trimer, quercetin, and protocatechuic acids”.

  1. More comparison of the obtained data with literary sources is desirable.

Answer: the new works were added in the results and discussion

Reviewer 2 Report

The article "Profile of Bioactive components of Cocoa (Theobroma ca-2 cao L.) By-products from Ecuador and Evaluation of their 3 Antioxidant Activity" by Wilma Llerena et al. follows the classic model for this type of material (Research Article), comprising four parts: Introduction, Materials and Methods, Results and Discussion, and Conclusions. The list of bibliographic references is adequate; the documentation regarding the titles consulted is appropriate.

I have some comments from the authors.

- I advise authors to always give the synonym of acronyms when they appear for the first time in the manuscript's text. Please review the entire manuscript.

- Line 150 - I suggest the authors present a flowchart describing sub-point 2.2 - "Cocoa Fermentation Process".

- Line 347-349 - Regarding the sentence "The CCN-51 cocoa pods have an average weight of 814 g, 27 g of placenta, 226 g of fresh almonds", how did you obtain the value of 814 grams? And where do "fresh almonds" come from, and what part of the process does it refer to?

- Line 345 - I suggest the authors review the units of the different results in this chapter. Please explain why to show some results in "g/100ml" and others in "g/100g". If possible, standardize the units.

   - I suggest the authors present a table summarizing the lines of work.

 Moderate editing of English language required

Author Response

Reviewer # 2:

Comments and Suggestions for Authors

The article "Profile of Bioactive components of Cocoa (Theobroma cacao L.) By-products from Ecuador and Evaluation of their Antioxidant Activity" by Wilma Llerena et al. follows the classic model for this type of material (Research Article), comprising four parts: Introduction, Materials and Methods, Results and Discussion, and Conclusions. The list of bibliographic references is adequate; the documentation regarding the titles consulted is appropriate.

I have some comments from the authors.

- I advise authors to always give the synonym of acronyms when they appear for the first time in the manuscript's text. Please review the entire manuscript.

Answer: Thanks, the abbreviations were reviewer in the manuscript.

- Line 150 - I suggest the authors present a flowchart describing sub-point 2.2 - "Cocoa Fermentation Process".

Answer: flowchart was added in the sub-point 2.2.

- Line 347-349 - Regarding the sentence "The CCN-51 cocoa pods have an average weight of 814 g, 27 g of placenta, 226 g of fresh almonds", how did you obtain the value of 814 grams? And where do "fresh almonds" come from, and what part of the process does it refer to?

Answer: This sentence was modified. “Each cocoa fruit of the CCN51 variety was previously weighed. CCN-51 cocoa pods present an average weight of 814 g, 27 g of placenta, 226 g of fresh almonds and each cob presented a number of approximately 49 grains. Figure 1 shows the percentages of cocoa residues in the different parts of the fruit”.

- Line 345 - I suggest the authors review the units of the different results in this chapter. Please explain why to show some results in "g/100ml" and others in "g/100g". If possible, standardize the units.

Answer: the mucilage units were presented in mg GAE/100 mL because this residue is very difficult to freeze-dry. Its viscosity changes in the freeze-drying process and it retains a lot of moisture. In the end, work was done on a fresh base.

    - I suggest the authors present a table summarizing the lines of work.

Answer: the flowchart was added. Figure 1.

Reviewer 3 Report

The manuscript entitled “Profile of Bioactive components of Cocoa (Theobroma cacao L.) By-products from Ecuador and Evaluation of their Antioxidant Activity” of Wilman Carrillo and his co-workers brings the study of the profile of bioactive compounds in the by-products of two cocoa varieties (Nacional x Trinitario type and CCN-51). Additionally, the authors discovered that the mucilage and the bean shell as by-products in cocoa production show a strong antioxidant activity. Consequently, the authors propose to use these by-products as nutritional ingredients in the food industry.

Although the study itself is not very genuine, it still contributes new knowledge that can help in more rational use of by-product in cocoa industry. Nevertheless, there are some points that need further clarification.

1.) In lines 407 and 408 it is written: “The extracts were obtained with acetone, ethanol, methanol and water. TPC values of 4182 were found; 2336; 2515; 1715mg GAE/100g DW”. The second sentence is not clear completely to me. I assume that the numbers 4182, 2336, 2515 and 1715 are referring to TPC expressed in mg of GAE/100g DW measured for the bioactive material extracted using acetone, ethanol, methanol and water, respectively. Please, clarify.

2.) In lines 424, 552, 583, and 612 “test student” is mentioned. As far as I understand, Student's t-statistical test is meant there (mentioned also in line 337 – however, considering that “Student” was Gosset’s pseudonym, the word “student” should be then written as “Student”). Nevertheless, the term “test student” has different meaning and it should not be used (neither in the form of “test student” nor in the form of “test Student”).

3.) In Tables 2, 3, and 4 results (content, antioxidant activity) are given per volume of mucilage and per weight of bean shell (by the way: for this last one – is it always DW meant?). Nevertheless, in Table 1 there is some confusion how the results are given – they are again given pre volume of mucilage and per weight of bean shell (or perhaps in mg GAE/100 g DW of bean shell and in mg CE/100 mL of bean shell). Further, it is not clear also what is exactly meant with “Statistical differences were indicated with superscript letters.” (what is meaning of a and what is meaning of b?).

4.) In lines 322 and 323 it is written “the team adds”. What is meant by this?

5.) Analytical errors given are sometime absurdly low. In Table 4, for example, it is written that the content of CAT in bean shell for Nacional x Trinitario type was determined to be “1616.62 ± 0.0001” (mg/100 g). That means that the relative error was 0.0001/1616.12 = 6e-8. There are just a few physical constants that can be determined with such accuracy but no concentration of chemical compounds. Note also that the last significant figure in the final result itself should be of the same order of magnitude (in the same decimal position) as the uncertainty.

6.) In the text there is not explicitly mentioned that “TE” means “Trolox equivalent” unit. Add this explanation, please.

7.) Are antioxidant activities of bean shell really so high as reported in Table 4 (tens of g TE/g) or it is a lapsus?

Minor remarks:

There are many typos in the manuscript:

-        double punctuation marks at the end of some references (e.g. [5], [17]) and sentences

-        references often end with a comma and not with a full stop

-        “And” in line 362

-        “of the profile. phenolic identified in exuded mucilage” in line 642

-        “).” in line 616

-        “(not determinate)” in line 657)

Not all the typos are mentioned here – read the manuscript again, please, and correct them.

The manuscript is in general written in an understandable mode although there exist also passages where some minor improvements of the language would be helpful.

Author Response

Reviewer # 3:

Comments and Suggestions for Authors

The manuscript entitled “Profile of Bioactive components of Cocoa (Theobroma cacao L.) By-products from Ecuador and Evaluation of their Antioxidant Activity” of Wilman Carrillo and his co-workers brings the study of the profile of bioactive compounds in the by-products of two cocoa varieties (Nacional x Trinitario type and CCN-51). Additionally, the authors discovered that the mucilage and the bean shell as by-products in cocoa production show a strong antioxidant activity. Consequently, the authors propose to use these by-products as nutritional ingredients in the food industry.

Although the study itself is not very genuine, it still contributes new knowledge that can help in more rational use of by-product in cocoa industry. Nevertheless, there are some points that need further clarification.

1.) In lines 407 and 408 it is written: “The extracts were obtained with acetone, ethanol, methanol and water. TPC values of 4182 were found; 2336; 2515; 1715mg GAE/100g DW”. The second sentence is not clear completely to me. I assume that the numbers 4182, 2336, 2515 and 1715 are referring to TPC expressed in mg of GAE/100g DW measured for the bioactive material extracted using acetone, ethanol, methanol and water, respectively. Please, clarify.

Answer: this paragraph was rewrite. Nsor-Atindana et al. [40] have described TPC and TFC content in cocoa bean shell extracts. The extracts were prepared with different solvents as acetone, ethanol, and methanol. TPC values reported were of range of 1715 - 4182 mg GAE/100g DW.

2.) In lines 424, 552, 583, and 612 “test student” is mentioned. As far as I understand, Student's t-statistical test is meant there (mentioned also in line 337 – however, considering that “Student” was Gosset’s pseudonym, the word “student” should be then written as “Student”). Nevertheless, the term “test student” has different meaning and it should not be used (neither in the form of “test student” nor in the form of “test Student”).

Answer: this error was modified for “test Student”

3.) In Tables 2, 3, and 4 results (content, antioxidant activity) are given per volume of mucilage and per weight of bean shell (by the way: for this last one – is it always DW meant?). Nevertheless, in Table 1 there is some confusion how the results are given – they are again given pre volume of mucilage and per weight of bean shell (or perhaps in mg GAE/100 g DW of bean shell and in mg CE/100 mL of bean shell). Further, it is not clear also what is exactly meant with “Statistical differences were indicated with superscript letters.” (what is meaning of a and what is meaning of b?).

Answer: the mucilage units were presented in mg GAE/100 mL because this residue is very difficult to freeze-dry. Its viscosity changes in the freeze-drying process and it retains a lot of moisture. In the end, work was done on a fresh base.

Statistical differences were indicated with superscript letters.” Answer: Results were expressed as mean ± standard deviation (SD) (n=9) and were analyzed by test Student. Statistical differences (p<0.05) were indicated with superscript letters. GAE (gallic acid equivalents) and CE (catechin equivalents). * Cocoa Bean shell was expresses as (mg GAE/100 g, DW).

4.) In lines 322 and 323 it is written “the team adds”. What is meant by this?

Answer: this sentence was rewrite: After, 150 μL of fluorescein (17.7 nM) were add, and was incubated at 37°C for 30 min. After, 25 μL of the AAPH solution (125.92 mM) were added.

5.) Analytical errors given are sometime absurdly low. In Table 4, for example, it is written that the content of CAT in bean shell for Nacional x Trinitario type was determined to be “1616.62 ± 0.0001” (mg/100 g). That means that the relative error was 0.0001/1616.12 = 6e-8. There are just a few physical constants that can be determined with such accuracy but no concentration of chemical compounds. Note also that the last significant figure in the final result itself should be of the same order of magnitude (in the same decimal position) as the uncertainty.

Answer: this error was corrected. 1616.62 ±0.01. Was an error of writing

6.) In the text there is not explicitly mentioned that “TE” means “Trolox equivalent” unit. Add this explanation, please.

This abbreviation was defined in the abstract: The antioxidant activity values (ABTS, FRAP and ORAC methods) were higher in the samples of Nacional x Trinitario type than CCN-51. The mucilage samples presented antioxidant values of (4.69; 3.35 and 1.28 µM trolox equivalents (TE)/mL) respectively and the bean shell samples presented antioxidant values of (8.54; 7.89 and 1.33 µM TE/mL) respectively.

Also, was defined in material and methods: The antioxidant activity was quantified by UV-VIS spectrophotometry (Shimadzu 2200 Spectrophotometer; Tokyo, Japan) against a Trolox standard curve (0-800 µmol/L). The curve obtained was (y = 0.0013 x + 0.126; R2 = 0.9982). All assays were performed in triplicate (n = 9) for 3 days. The results were expressed as µM of trolox equivalents (TE)/mL of sample.

This abbreviation (TE) was defined in the table 4

7.) Are antioxidant activities of bean shell really so high as reported in Table 4 (tens of g TE/g) or it is a lapsus?

 Answer: This error was corrected. The correct unit was mg TE/g is correct. This mistake was corrected in the Table 4 and the text. Thanks for the observation.

Minor remarks:

There are many typos in the manuscript:

-        double punctuation marks at the end of some references (e.g. [5], [17]) and sentences

-        references often end with a comma and not with a full stop

Answer: this mistake was corrected in the manuscript

-        “And” in line 362

Answer: this mistake was corrected for: Figueroa et al. [34] and Campos-Vega et

-        “of the profile. phenolic identified in exuded mucilage” in line 642

Answer: this mistake was corrected for: that are part of the profile phenolic

-        “).” in line 616

(8.54; 7.89 and 1.33 µM TE/mL).

-        “(not determinate)” in line 657)

Answer: this mistake was corrected for not determined

Not all the typos are mentioned here – read the manuscript again, please, and correct them.

Reviewer 4 Report

The paper with the title Profile of bioactive components of cocoa byproducts (Theobroma cacao L.) from Ecuador and evaluation of their antioxidant activity revealed the presence of bioactive compounds in the external parts of cocoa beans considered cocoa byproducts. HPLC analysis allowed the identification and quantification of the phytonutrient profile of cocoa mucilage and bean shell and methylxanthines in the two cultivars grown in Ecuador. The secondary products resulting from the processing of the two varieties of cocoa showed a strong antioxidant activity. Thus, cocoa residues have added value and can be used in the future for the formulation of functional ingredients in the food industry. 

Author Response

Answer: Thanks for your comments about the work

Reviewer 5 Report

Profile of Bioactive components of Cocoa By-products in two cultivars and their antioxidant activities were presented. The topic is interesting. However, many issues should be addressed before it can be forwarded to further processing.

1. The redundant descriptions are not necessary, such as “Reporting values of 171.32 g TE/g in ABTS, 192.22 g TE/g for FRAP and 56.87 619 g TE/g in ORAC for CCN-51 samples. In the Nacional x Trinitario type cocoa, the values 620 were 167.06 g TE/g for ABTS, 16.01 g TE/g in FRAP and 52.53 g TE/g in ORAC.(Lines 619-621)

2.The data should be carefully checked, such as those in Lines 614-616.

3.About measuring units, the unit of measurement of the correlation coefficient (%) in the second Table 4 remains to be discussed.

4. The authors were very careless in writing the manuscript and the text should be checked carefully. There were two Table 4 in the context.

5.About the grammar, some sentences are not complete, such those in Lines 619-620.

6.Preparation of plant samples (including fermentation) has great impact on the phytochemical composition and levels. In this manuscript, the fermentation in Nacional x Trinitario type cacao was carried out for 4 days. However, the fermentation of CCN-51 cocoa was carried out for 6 days. It can ben considered that the concentrations snd antioxidant activity of bioactive components between these two cultivars were from the difference in fermentation, instead of cultivars. In this case, the conclusions of the manuscript are questionable.

Author Response

Reviewer # 5:

Comments and Suggestions for Authors

Profile of Bioactive components of Cocoa By-products in two cultivars and their antioxidant activities were presented. The topic is interesting. However, many issues should be addressed before it can be forwarded to further processing.

  1. The redundant descriptions are not necessary, such as “Reporting values of 171.32 g TE/g in ABTS, 192.22 g TE/g for FRAP and 56.87 619 g TE/g in ORAC for CCN-51 samples. In the Nacional x Trinitario type cocoa, the values 620 were 167.06 g TE/g for ABTS, 16.01 g TE/g in FRAP and 52.53 g TE/g in ORAC. (Lines 619-621)

2.The data should be carefully checked, such as those in Lines 614-616.

According to the results obtained, the cocoa mucilage from the CCN-51 variety presented lower values ​​of antioxidant activity (4.69; 3.35 and 1.28 µM TE/mL) than the National x Trinitario type cocoa samples (8.54; 7.89 and 1.33 µM TE/mL).

3.About measuring units, the unit of measurement of the correlation coefficient (%) in the second Table 4 remains to be discussed.

Answer: For convenience we express it as a percentage (%) but the value is dimensionless. The correct way is without the percentage. We have corrected it in the table and in the text of the manuscript. Thanks for the observation.

Correlation less than zero: If the correlation is less than zero, it means that it is negative, that is, that the variables are inversely related.

When the value of one variable is high, the value of the other variable is low. The closer it is to -1, the clearer the extreme covariation. If the coefficient is equal to -1, we refer to a perfect negative correlation.

Correlation greater than zero: If the correlation is equal to +1 it means that it is perfect positive. In this case it means that the correlation is positive, that is, that the variables are directly correlated.

When the value of one variable is high, the value of the other is also high, the same thing happens when they are low. If it is close to +1, the coefficient will be the covariation.

  1. The authors were very careless in writing the manuscript and the text should be checked carefully. There were two Table 4 in the context.

Answer: this error was corrected. Table 4 and Table 5

5.About the grammar, some sentences are not complete, such those in Lines 619-620.

Answer: this sentence was rewrite

6.Preparation of plant samples (including fermentation) has great impact on the phytochemical composition and levels. In this manuscript, the fermentation in Nacional x Trinitario type cacao was carried out for 4 days. However, the fermentation of CCN-51 cocoa was carried out for 6 days. It can ben considered that the concentrations and antioxidant activity of bioactive components between these two cultivars were from the difference in fermentation, instead of cultivars. In this case, the conclusions of the manuscript are questionable.

Answer: The mucilage exuded from cocoa, before fermentation, is a white viscous substance that adheres to and surrounds the beans; It represents between 3.50 (National*Trinitario) and 5.30% (CCN-51) of the total weight of the cocoa fruit, depending on the variety of origin. During fermentation, the pulp or mucilage is removed and hydrolyzed by microorganisms, in a period of 4 days in materials with a lower percentage of mucilage and 7 days in foreign materials with a higher pulp content. This process is necessary for the production of alcohol and acetic acid in almond fermentation, therefore, the hydrolyzed pulp drains as exudate, involving various microbial colonies throughout the process; initially they are dominated by yeasts and later by lactic acid bacteria.

This microbial activity generates metabolites and conditions that decrease the growth viability of the almond embryo, triggering a series of biochemical reactions and chemical changes that are essential for the development of the complex “chocolate” flavor. To define the fermentation times, in each material the level of acidity (acetic flavor) of the grain is considered with the level of fermentation. Incomplete fermentation results in a lack of chocolate flavor (vinegared cocoa), while off-flavors result from excessive fermentation and spoilage of the almonds. In the cocoa sector, the fermentation time of each material has been widely established, so it is known that fine type cocoa almonds (Nacional xTrinitario) require a short fermentation (4 days). On the contrary, Forasteros cocoa almonds such as CCN-51 need approximately 6-7 days to reduce their acidity and bitterness. When fresh, cocoa beans have a high content of polyphenols (15-20% in fresh seeds without fat), which provide an astringent, bitter and unpleasant flavor which should be reduced to 5%. The optimal point of fermentation is determined through a parameter called fermentation index which relates the content of brown products and flavonoid condensate (460 nm) and anthocyanins (525 nm) must be equal to 1; indicating that the oxidation and condensation reactions have finished. In Nacionalx Trinitario cacao this process is completed after 4 days of fermentation and in CCN-51 cacao after 6 days of fermentation.

Bioactive compounds (antioxidants) generally accumulate in the different structures of cocoa fruits (shells, pods, mucilage and placenta); as well as in different parts of the plant such as leaves, flowers, seeds and stems. However, the objective of this work is not to compare the phytochemical composition of the edible and inedible parts of the Theobroma Cacao, L. plant, but to value the residues that are currently produced in the country by the cocoa activity under the established conditions. through the agricultural production chain. This potential raw material for obtaining antioxidants is characterized by being cheap, renewable and abundant; But in Ecuador, two cocoa materials are mainly produced; the fine type (National x Trinitario type) and in bulk (CCN-51); The industrial sector has already established the production conditions for each material, for which this research adopted the industry conditions and compared the bioactive contents of the two materials, which will allow agro-valorization of the waste produced and contribute to knowledge. of its phytochemical composition.

Noor Ariefandie Febrianto, Sunan Wang & Fan Zhu (2022) Chemical and biological properties of cocoa beans affected by processing: a review, Critical Reviews in Food Science and Nutrition, 62:30, 8403-8434, DOI: 10.1080/10408398.2021.1928597

“The majority of cocoa beans available in the market are processed by means of conventional method (Figure 1). In this report, conventional processing refers to the commonly applied methods by farmers or cocoa industries (Figure 1B), while non-conventional processing (Figure 1A,C) relates to alternative methods or has additional steps of operations added to the conventional processing. In conventional processing, freshly harvested cocoa beans are immediately subjected to spontaneous fermentation for 2–7 days (depending on the cocoa varieties, regions of growth and common practices) before drying (Urbanska et al.-2019; Munoz et al. 2020)”.

Round 2

Reviewer 3 Report

The authors have addressed majority of comments that I gave during the first stage of the review. Nevertheless, their response seems to me inadequate in few points:

1.) One of my remarks was about admission of errors in HPLC analysis. As the most striking example I mentioned the extremely low error reported for the content of catechin in Cocoa bean shell.

Upon my remark, the authors have changed the error of quantification of catechin in Cocoa bean shell (Nacional x Trinitario type) from ± 0.0001 to ± 0.01 (mg/100 g). Therefore, I ask authors again more explicitly:

a) What was the quality of the catechin standard? How the uncertainty in the content of the catechin in the catechin standard is incorporated in the error of the content of catechin reported in Table 2?

b) How precisely you can measure absorbance with the HPLC diode array detector? How is the precision of the detector incorporated in the error of the content of catechin reported in Table 2?

c) How did the authors convert surface of the peak (Figure 3d) of catechin into catechin concentration? Are the authors really capable to set the baseline so reliably to integrate, for example, peak of catechin on chromatogram (Fig. 3c) with the relative error of 0.02/499.45 = 0.00004 (taken from Table 2) or even (Fig. 3d) with the relative error of 0.01/1616.62 = 0.000006 (also taken from Table 2)?

d) Why the peaks of caffeine in Nacional x Trinitario type variety are so differently integrated (chromatograms in Fig. 4b [Cocoa mucilage] and in Fig. 4d [Cocoa bean shell]) and still the relative error in the content of caffeine in Cocoa bean shell (Table 3) is claimed to be 0.001/0.36 = 0.003)?

I have been working for some time with HPLC instrumentation in the research department of one pharmaceutical company and I find the upper mentioned figures as a joke. If the authors are not analytical chemists, then they should ask analytical chemists how large the real error is.

Considering that I put forward a request to quote errors realistically already during my first review, but the authors more or less ignored my remark, I will be a bit rude now and I will repeat my request again in capital letters:

DEAR AUTHORS, CHECK ALL THE ERRORS YOU ARE QUOTING IN YOUR MANUSCRIPT (NOT ONLY FOR CATECHIN) AND DO YOUR WORK PROPERLY. REPEAT STUDENT STATISTIC WITH NEWLY ESTIMATED ERRORS.

Note again that the last significant figure reported in the result itself should be in principle placed in the same decimal position as the order of magnitude of the absolute error.

2.) I ask authors again to:

a) capitalize »student's« to »Student's« (line 306)

b) change »test Student« to »Student's t-test« (lines 449, 594, 624, 653)

Author Response

Answer (1a, b, c, and d): The HPLC used for the analysis was HPLC (Agilent technologies series 1100/1200; Waldbronn, Germany), with the help of a DAD detector at a wavelength of 273 nm according to the method described by Samaniego et al. [5], the separation was performed using an Agilent Zorbax SB C18 column (150 mm x 4.6 mm; 5 μm particle size). It does indeed have a DAD detector. The wavelength was adjusted according to the analyte to be characterized. Mucilage by-products samples are difficult to work with, for this reason there are few works on mucilage. Analyzing those samples by HPLC was quite complicated and the baseline is not perfect because it is also an extract. The extracts are not pure standards, they are mixtures of compounds that are isolated to the maximum for analysis but may have other compounds that may interfere. The concentrations of the phenolic compounds and methylxanthines were calculated by the team using the calibration curves of the Fig S1 and Fig S2 standards. The data of the standard deviations were reviewed and where there was an error in the transcription of the data it was corrected. The data are the mean of the concentration of three samples that were analyzed and were prepared under the same conditions and are from the same cocoa crop. Data (±) are the standard deviation (SD) of the means of those three measured concentrations. The values of the three concentrations were very similar, which is why the values of their standard deviations are low. The chromatograms as a whole, for dealing with complex samples of by-products, were quite acceptable. 

2.) I ask authors again to:

  1. a) capitalize »student's« to »Student's« (line 306)
  2. b) change »test Student« to »Student's t-test« (lines 449, 594, 624, 653)

Answer: This was change for Student’s t-Test

Reviewer 5 Report

The concerned issues were addressed.

Author Response

Thanks for your comments.